# A Comparison of Different Counting Methods for a Holographic Particle Counter: Designs, Validations and Results

**DOI:** 10.3390/s20103006

**Published:** 2020-05-25

**Authors:** Georg Brunnhofer, Isabella Hinterleitner, Alexander Bergmann, Martin Kraft

**Affiliations:** 1Nanophysics & Sensor Technologies, AVL List GmbH, 8020 Graz, Austria; 2Sensor Systems, Silicon Austria Labs GmbH, 9524 Villach/St. Magdalen, Austria; hinterleitner@techmeetslegal.at (I.H.); martin.kraft@silicon-austria.com (M.K.); 3Institute of Electronic Sensor Systems, Graz University of Technology, 8010 Graz, Austria; alexander.bergmann@tugraz.at

**Keywords:** holographic particle counter, circular Hough transform (CHT), blob detection, deep convolutional neural network

## Abstract

Digital Inline Holography (DIH) is used in many fields of Three-Dimensional (3D) imaging to locate micro or nano-particles in a volume and determine their size, shape or trajectories. A variety of different wavefront reconstruction approaches have been developed for 3D profiling and tracking to study particles’ morphology or visualize flow fields. The novel application of Holographic Particle Counters (HPCs) requires observing particle densities in a given sampling volume which does not strictly necessitate the reconstruction of particles. Such typically spherical objects yield circular intereference patterns—also referred to as fringe patterns—at the hologram plane which can be detected by simpler Two-Dimensional (2D) image processing means. The determination of particle number concentrations (number of particles/unit volume [#/cm${}^{3}$]) may therefore be based on the counting of fringe patterns at the hologram plane. In this work, we explain the nature of fringe patterns and extract the most relevant features provided at the hologram plane. The features aid the identification and selection of suitable pattern recognition techniques and its parameterization. We then present three different techniques which are customized for the detection and counting of fringe patterns and compare them in terms of detection performance and computational speed.

## 1. Introduction

In the field of aerosol measurement, the determination of particle number concentrations is a major topic. Optical Particle Counters (OPCs) represent the most widespread instruments for obtaining Particle Number (PN), based on the optical detection of spatially separated particles. Recently, Ref. [1] introduced a Holographic Particle Counter to image a certain sampling volume and count all present particles at once. Because of the holographic approach, imaged particles are recognized as interference patterns. In typical holographic particle imaging applications, 3D reconstruction algorithms are used to reconstruct the particles in the sampled volume [2,3,4,5,6] for the detection, characterization and visualization of the particles’ fields. OPCs primarily aim to count aerosol particles wherefore wavefront reconstruction is not strictly necessary. The tracing of particles and the determination of their size is mostly an ancillary indicator of proper device operation or maintenance need. Under these aspects, the recognition of particles as valid interference patterns at the Two-Dimensional hologram plane is sufficient and can be performed with common pattern recognition techniques.

The herein presented work addresses the design, evaluation and comparison of different pattern recognition techniques to detect and count particles. To validate the performance of the proposed recognition techniques on real world measurement data, the Particle Imaging Unit (PIU) developed in [1] is used, which is the primary application of the presented counting methods. It resolves particles that are larger than roughly 3–4 μm which corresponds to pixel size of the imager. It is operated in the same setup as outlined in their work where the imaging unit is set on top of a so called Condensation Nucleus Magnifier (CNM) which grows particles to a homogeneous and predetermined size of around $d_{prt}=7\;\mu m$ by means of condensation (particles are condensed by a working fluid—n-decane in this case—to form individual droplets). Subsequently, these droplets are imaged in the PIU and yield the particles’ interference patterns. To validate the detection performance of the presented pattern recognition methods, a referencing Condensation Particle Counter (CPC) (A CPC is based on the same working principle but with a different optical counting approach.) is taken for comparison. CPCs output particle counting rates in terms of particle number concentrations in numberofparticles/unitvolume = [#/cm${}^{3}$]. Because the generation and supply of particles at an unambiguous, continuous and reproducible rate is practically impossible, a comparison of number concentrations is most reasonable. In Figure 1, the PIU is sketched on the right side and its working principle shown on the left.

Particles in the sampling cell of the PIU are illuminated by a reference plane wave, generated by a low coherence diode laser. Each particle in the sampling channel acts as a single point-like object which diffracts the incident plane wave to yield a spherical object wave. Both wave fronts propagate along the *z*-axis and interfere in a distance $z_{prt}$ at the detection- or hologram plane.

## 2. Fringe Patterns and Its Features

When talking about point-like objects such as a particle in Figure 1, the interference pattern is also referred to as a fringe pattern which is a set of radially symmetric rings or fringes. Constructive and destructive interference lead to bright and dark fringes at the detection plane whereby the depth information of the object is carried by the phase information of the pattern.

### 2.1. Information Content of Fringe Patterns

The fringe pattern of one particle is described by the following function [3,7]:(1)$$\Psi_{prt}\left(x,y\right)=C_{1}+C_{2}\cdot sin\left(\frac{\pi }{\lambda \cdot z_{prt}}\left(x^{2}+y^{2}\right)\right)$$
where $C_{1}$ is a constant bias summarizing the intensities of the reference wave and the object wave. The change of constructive and destructive interference is given by a sine-function with amplitude $C_{2}$. The phase information yields the density of fringes—more precisely the spatial rate of change—which linearly increases along the xy-plane from the center to the outside of the pattern. This spatial rate of change is known as the fringe frequency [7] with:(2)$$\nu =\frac{1}{2\pi }\frac{d}{dx}\left(\frac{\pi }{\lambda \cdot z_{prt}}x^{2}\right)=\frac{x}{\lambda \cdot z_{prt}}$$
where *x* is the spatial coordinate, λ the wavelength of illumination and $z_{prt}$ the position of the particle along the illumination path. That key characteristic of the spatial frequency increase is also known in digital image processing to test filter approaches [8] and will be made use of later in this report. The radii of fringes and, thus, the extent of a fringe pattern at the detection plane depends on $z_{prt}$: particles in a larger distance $z_{prt}$ yield large fringe patterns and close particles lead to smallers. The range of possible *z*-distances is determined by the depth of the sampling volume (here the sampling channel depth $z_{ch}$ of the PIU in Figure 1) and are a reason for a certain size distribution of fringe patterns at the detection plane, depicted in Figure 2a. On the right side, the normalized intensity histogram is illustrated. The question to address is now what are the smallest and largest possible sizes of patterns which need to be detected.

### 2.2. Features to Extract

The formation of fringe patterns can be described by the Angular Spectrum Method (ASM) as in [9]. A fringe pattern, however, also largely complies with a sinusoidal Fresnel Zone Plate (FZP) [7] which is primarily used to focus light, based on diffraction. The spacing of zones is such that light transmitted by the transparent zones constructively interferes at a desired focus. Holograms follow the same principle, i.e., its interference rings are caused by a common focus point—the particle. The idea here is to make use of the analogy to zone plates to easily estimate the extent of fringe patterns. As the focal length *f* in Figure 3 corresponds to the distance $z_{prt}$ from the particle to the detection plane, each zone *n* may be interpreted as a fringe of radius [10]:(3)$$R_{n}≃\sqrt{n\cdot \lambda \cdot z_{prt}}$$
with λ the wavelength of the illumination light. At the detection plane, constructive interferences appear as bright fringes and correlate to zones of even multiples of *n* in a zone plate.

The smallest circle radius $R_{min}$ to be expected with the utilized sampling cell is therefore estimated by calculating Equation (Equation 3) for the first even zone n=2 and at the smallest possible distance $z_{0}$ (see Figure 1 right). Conversely, the largest circle radius $R_{max}$ is obtained with the largest resolvable zone $n_{max}$ at the furthermost possible particle location $z_{prt}=z_{0}+z_{ch}$, where $z_{ch}$ in this case is the sampling channel depth of the PIU and, thus, the furthermost boundary. With that given Depth of Field (DoF) in a range of $\left\{z_{0},\;\ldots \;,z_{0}+z_{ch}\right\}$, the smallest circle radius to recognize is found with $R_{min}=19\;pxl$ and the largest with $R_{max}=70\;pxl$, where the outermost meaningful fringe is considered with n=10. One practicable goal is now to recognize patterns which consist of a set of concentric circles of radii in the range of R=19…70pxl.

Another beneficial circumstance provided by the given DoF is that center zones are predominately dark blobs as apparent from Figure 2. Instead of detecting circles as fringes, center zones may be targeted as well for the recognition of fringe patterns.

### 2.3. Intensity Dependence of Fringe Patterns

The intensity of diffracted waves is magnitudes lower than the direct beam, which lead to fringe patterns of a fairly low contrast. Besides the illumination intensity incident onto particles, the contrast is additionally affected by a variety of influences, such as the particle size and diffraction properties of particles and multiple scattering as a function of particle concentrations [1,11]. As a consequence, a wide span of contrast adds additional difficulty to the feature extraction and its parameterization of optimal sensitivity.

## 3. Methods

### 3.1. Customized Hough Transform

The Hough Transform (HT) is a very well known feature extraction technique in digital image processing for detecting arbitrary geometrical shapes such as straight lines, circles or ellipses [12]. It makes use of a parameter space—the so-called Accumulator or Hough Space—where a voting procedure is carried out over a set of parameterized image objects; here circles with a certain range of radii. Object edge points, which ultimately form the object’s shape, are transformed into that parameter space using its respective mathematical representation. The resulting accumulated feature candidates allow for easier grouping and are therefore robust in the presence of noise, occlusions and varying illuminations.

#### 3.1.1. Working Principle

The implemented customized HT is based on the work of Atherton and Kerbyson [13,14] where the edge filtered image is convolved with a complex filter operator:$$\begin{array}{c} O_{PCA}\left(x,y\right)=\left\{\begin{array}{cc} e^{j\cdot \varphi_{xy}} & \mathrm{iff}\;R_{min}^{2}<x^{2}+y^{2}<R_{max}^{2}\\ 0 & \mathrm{otherwise} \end{array}\right. \end{array}$$
that forms a Phase Coded Annulus. In this manner, the range of scanned circle radii between $R_{min}$ and $R_{max}$ is phase-coded (from 0 to 2π) into a complex accumulator space with the phase coding across the annulus following a log coding:(4)$$\varphi_{xy}=2\pi \left(\frac{log\left[\sqrt{\left(x^{2}+y^{2}\right)}\right]-logR_{min}}{logR_{max}-logR_{min}}\right)$$

In parameter space, constructive accumulation now occurs only at bins where the transformed candidates intersect with the same phase—the bin which corresponds to the circle’s center. Centers are then estimated by detecting such bins as peaks and determining its centroids using geometric moments (see also Section 3.2.3). The sensitivity of that peak detection is in the range of $S_{HT}=\left\{0..1\right\}$ and leads to fewer detected circles at lower sensitivity levels.

The radii are estimated by simply decoding the phase information from the estimated center location.

#### 3.1.2. Image Preprocessing

In order to enhance the Signal to Noise Ratio (SNR), and thus improve fringe visibility of the patterns prior to the edge filtering, a Gaussian smoothing kernel is taken. Since higher order fringes are not mandatory for the recognition of valid patterns, the filter size of the lowpass kernel and ultimately its standard deviation $\sigma_{lp}$ is configured to filter the unwanted outermost fringes. The cutoff frequency of the filter is determined by making use of one of the most common and heuristic measures when dealing with Gaussian distributions, known as the 3-sigma rule [15]. The smallest feature in an image unaffected by filtering has to fit within the 3σ or 99% confidence interval. In terms of a fringe pattern, the distance between fringes of the same parity (even to even or odd to odd) may be interpreted as the smallest detail to preserve. This distance Δdr in Figure 3 can be estimated as the total width of two successive zones (opposite parity) by making again use of the FZP from [16]:(5)$$\begin{array}{cc} \Delta dr≃\sum_{m=n}^{n+1}dr_{m} & =\sum_{m=n}^{n+1}\frac{\sqrt{m\cdot \lambda \cdot z_{prt}}}{2m} \end{array}$$
with
(6)$$\begin{array}{cc} dr_{n} & =\frac{R_{n}}{2n} \end{array}$$

Considering that the 99% confidence interval of a Gaussian filter kernel is its total width with $6\sigma_{lp}+1$, the filter size of a Gaussian lowpass filter can be calculated by:(7)$$\sigma_{lp}=\frac{\Delta dr-1}{6}$$

#### 3.1.3. Parameterization

From Equation (Equation 3), it is clear that the radii $R_{n}$ of fringes are only dependent on their respective fringe index *n* within the pattern and the distance $z_{prt}$ from the particle (the wavelength λ is constant). The range of possible distances $z_{prt}$ is bounded by the sampling channel. Taking into account that the innermost fringe is principally sufficient for pattern recognition, the range of radii to search may be truncated which greatly speeds up the HT. In this work, a single-step approach is chosen where only the innermost fringe at n=2 is searched with $R_{2}=22\ldots 33\;pxl$. Due to the geometries of the given sampling cell the innermost fringe may span a range of $R_{2}=23\ldots 32\;pxl$. A margin of ±1pxl is added. Analyses showed that especially at high particle densities the overlap of fringe patterns is too strong to identify higher order fringes. Moreover, its intensities tend to be too low to be detected. Thus, the Gaussian filter is set to a cutoff of $\sigma_{lp}=2.62$ to retain an approximate level of detail of Δdr≈17pxl. The sensitivity is set to $S_{HT}=0.93$ and was heuristically determined.

### 3.2. Blob Detection

Blob detection is a subcategory of image matching techniques, aiming to detect regions of common properties such as a homogeneous brightness or grayscale that thereby distinguish them from background regions [15,17,18,19]. Blob detectors can be based on image gradients (contrast), eigenvalues or templates [19]. Since the mathematical representation of fringe patterns is known, template matching [20] is a suitable approach.

#### 3.2.1. Blob Extraction Using Template Matching

An artificially generated fringe pattern is of course a viable template to use. However, since patterns start to overlap strongly at higher particle number concentrations, a mask that emphasizes the sole center zone is more meaningful. A multi-step template matching is performed using circular masks of steadily increasing radii $R_{m}$:$$\begin{array}{cc} g_{m}^{TM}\left(x,y\right) & =\left\{\begin{array}{cc} 1 & \mathrm{iff}\;{\left(x-\bar{x}\right)}^{2}+{\left(y-\bar{y}\right)}^{2}<=R_{m}^{2}\\ 0 & \mathrm{otherwise} \end{array}\right. \end{array}$$
with $\left(\bar{x},\bar{y}\right)$ the circle center and *m* the current step. The templates equal non-normalized disk-like box filter kernels that gradually lowpass-filter background noise with increasing radius of the masks and thereby emphasize regions that match it.

#### 3.2.2. Blob Segmentation

To segment blobs, global thresholding is necessary to find the optimal threshold in the histogram. Although Otsu’s method is one of the most widespread thresholding techniques due to its good performance and yet simplicity, it faces clustering problems with unimodal histograms. Small object areas compared to background areas are the cause for unimodality as reported in [21,22]. Unimodality however is the major case in our presented work and thus disqualifies methods of that kind of clustering thresholding. Instead, with the maximum entropy thresholding [23,24], an entropy-based thresholding method is utilized that interprets the maximum entropy as indicative of maximum information transfer.

It is based on the probability distribution function of gray-level histograms. Assuming two distributions where one belongs to the class of blobs (dark pixels) and the other to the class of background (bright pixels), then the optimum threshold $k_{opt}$ of inter-class entropy is found at:(8)$$k_{opt}=\underset{L_{1}\leq k<L_{2}}{\mathrm{argmax}}\left[H_{drk}\left(k\right)+H_{br}\left(k\right)\right]$$
where $H_{drk}$ is the entropy of dark pixels, based on the probability $P_{drk}$ that pixels are assigned to the class of dark pixels, and $H_{br}$ the entropy of bright pixels with its probability $P_{br}$, respectively. A lower limit $L_{1}$ and an upper limit $L_{2}$ confine the threshold $k_{opt}$ to a certain gray-level range for later discussed reasons. The standard setting is $L_{1}=0$ and $L_{2}=L$, where *L* is the number of gray-levels. With a set of k={0,1,2,…,L−1} of *L* gray-levels, the entropies of both classes are calculated as follows [23]:(9)$$H_{drk}\left(k\right)=-\sum_{l=0}^{k}P_{drk}\left(l\right)\cdot log\left[P_{drk}\left(l\right)\right]\;\mathrm{and}\;H_{br}\left(k\right)=-\sum_{l=k+1}^{L}P_{br}\left(l\right)\cdot log\left[P_{br}\left(l\right)\right]$$

#### 3.2.3. Blob Labeling and Counting

After thresholding, blobs remain as regions of connected pixels in the binary image and are typically detected by connected components labeling [25]. All connected or neighboring pixels corresponding to a separate region are assigned the same labels. The total number of different labels equals the number of detected blobs and, in the ideal case, also equals the total number of fringe patterns. In fact, fringe patterns may strongly overlap and yield merged blobs though. In order to discover such scenarios, blob features are meaningful to assess using different descriptors.

Regional descriptors are very often used in combination with connected components labeling and are based on mathematical moments of the form [20]:(10)$$m_{pq}=\sum_{(x,y)\in R}x^{p}y^{q}\cdot I(x,y)$$
where (p,q) are the indices of the moment and (x,y) the pixels of the region R in gray-scale images I(x,y). The sum p+q of the indices corresponds to the order of the moment $m_{pq}$. For binary images, as given after thresholding, the term I(x,y) equals 1.

Moments carry physical interpretations of shapes such as the mass (area), center of mass or gravity (centroid), eccentricity or orientation of the region. Therein, the order of the moment determines the property. The most common are the zeroth order moment $m_{00}$ as the area *A* and the first order moment as the centroid with $\bar{x}=m_{10}/m_{00}$ and $\bar{y}=m_{01}/m_{00}$. The centroid is also a common feature to locate or tag regions at its center point. In the given problem statement, it is of particular significance because the centroid of fringes represents the actual location in the xy- plane (see in Figure 1). In conjunction with the perimeter *P* (a boundary descriptor), the circularity is another meaningful descriptor [8]:(11)$$circularity=\frac{4\pi A}{P^{2}}$$

It is a measure independent of size, orientation, and translation, and is 1 for a circle. Merged blobs form elongated or asymmetric shapes that deviate strongly from the ideal circularity of 1 and therefore indicate multiple fringe patterns. In this work, it is used as a correction means which adds an additional count to regions where the circularity is beneath a threshold of circularity≤0.95.

### 3.3. Deep Convolutional Neural Network (DCNN)

Convolutional neural networks have made some great advances in visual recognition tasks, e.g., [26]. While convolutional neural networks have been used for a long time [27], their success was limited due to the size of available training sets and the size of available networks. A breakthrough has been achieved by Krizhevsky et al. [28] who were able to supervise a training of a large network with eight layers and millions of parameters on the ImageNet dataset with 1 million training images. Since then, even larger and deeper networks have been trained [29].

#### 3.3.1. Working Principle

The network architecture is depicted in Figure 4 and is based on the principle of a U-Net structure [30]. In total, the network has 23 convolutional layers. It comprises a contracting information path (left path) and an expansive path (right path). The contracting path follows the architecture of a convolutional network and includes the successive application of two 3 × 3 convolutions, each followed by a Rectified Linear Unit (ReLu) for activation and a 2 × 2 max pooling operation for downsampling. At each downsampling step, the number of feature channels is doubled. Every step in the expansive path consists of an upsampling of the feature map followed by a 2 × 2 convolution, a concatenation with a correspondingly cropped feature map from the contracting path, and two 3 × 3 convolutions, followed by a ReLu. Cropping is required due to the loss of boarder pixels at every convolution.

#### 3.3.2. Training

Input images and its corresponding segmentation maps are used to train the network with the stochastic gradient descent implementation of [31]. Due to the unpadded convolutions, the output image is smaller than the input by a constant border width. To minimize overhead and make maximum use of the Graphics Processing Unit (GPU) memory, large inputs are favored over a large batch size. Hence, the batch is reduced to a single image. Accordingly, a high momentum (0.99) is used, such that a large number of the previously seen training samples determine the update in the current optimization step.

#### 3.3.3. Data Processing and Evaluation

Data augmentation is the main process to teach the network the desired invariance and robustness characteristics in case only a few training samples are available. In case of fringe patterns, shift and rotation invariance are needed as well as robustness to deformations and gray value variations. The data set is provided by the EM segmentation challenge [32] that was started at ISBI 2012 and is still open for new contributions. The training data are a set of 30 frames (512 × 512 pixels) from the challenge. Each image within this data set is delivered with a corresponding fully annotated ground truth segmentation map for cells (white) and other structures within this challenge (black). In a second step, artificial data generated with the Aerosol Particle Model (APM) from Brunnhofer and Bergmann [9] was trained. The network is finally trained on real world measurement samples aside from simulated data sets.

An evaluation of the U-Net segmentation can be conducted by looking at the model accuracy and model loss for the training and validation set at hand (a data set as e.g., in Figure 2a).

## 4. Results

In the following section, the results of the customized HT, blob detection and the DCNN are compared in terms of detection performance and computational speed and which method is most suitable for the application in Holographic Particle Counters. For that purpose, an imaged hologram of a real measurement sample is taken as an example image where the density of fringe patterns is moderately high. Subsequently, a selected section of that image is used to first assess the HT and blob detection on an empirical basis. It contains strongly overlapping but also clearly separated fringe patterns and poses certain complexities to both methods. The neural network needs to be assessed with multiple validation images instead.

The second part outlines a qualitative comparison of all methods based on a real measured data set.

### 4.1. Customized HT

The aforementioned image section in Figure 5b shows mutliple overlaps of several fringe patterns in the lower right corner and a very strong merge in the lower left part. The dense group of patterns in the right corner is almost entirely detected except for one missing hit (tagged in orange). The preprocessed image (Figure 5a) reveals that all innermost fringes are resolved very sharply and suggests that the sensitivity $S_{HT}$ of the HT should be refined to recognize the missing hit as well. However, a higher sensitivity was identified to lead to an increase in false-positive hits and was hence deliberately avoided.

Strong overlaps in the lower left corner are reliably separated. While a distinction of the left two patterns can be validated through visual inspections and experiences, the right group is strongly bundled and requires backpropagation means to verify the particles in the reconstructed 3D-volume Not elaborated in this work, but the actual count of particles in this particular bundle is 4 indeed.

### 4.2. Blob Detection

The majority of the center spots in fringe patterns is dark. These dark centers equal multi-scale blobs that are extracted by the multi-step template matching approach. Since the filter kernels of the templates are non-normalized, a bias in pixel intensities is introduced during filtering. With respect to the image histogram, it narrows the mainlobe of Gaussian intensity distribution due to lowpass-filtering and shifts it to brighter intensity values as a consequence of the bias (compare the histograms of Figure 2b with Figure 6a).

The final histogram is right-sided with a mainlobe relating to the background and a certain side-distribution to its left which contains the information of the darker center blobs (and fringes of odd parity).

With maximum entropy thresholding, the optimal intensity threshold $k_{opt}$ is found where the best contrast is obtained in terms of maximum information transfer. Its indicative measure is the maximum sum entropy $H_{max}=\max\left[H_{br}\left(k\right)+H_{drk}\left(k\right)\right]$. Figure 6a shows the determination of the sum entropy with the entropy $H_{br}$ of bright pixels and the entropy $H_{drk}$ of dark pixels plotted separately. In this particular example image, the correct optimum threshold is located close to the left of the mainlobe at $k_{opt}=0.74$. However, the algorithm would actually fail to find that threshold because the background entropy $H_{br}$ gains a peak at lower pixel intensities (at k=0.37) and the actual maximum sum entropy would be erroneously reached at that particular threshold value. To avoid such misinterpretations, the thresholding limits $L_{1}$ and $L_{2}$ in Equation (8) are introduced. The upper limit $L_{2}$ equals the histogram bin of the mainlobe peak and is determined for each sample image individually. Evident from the given fringe pattern properties, the intensity of blobs will not exceed background levels and is therefore always located left of the mainlobe. $L_{1}$ is a rather empirical value and is set to 0.5 as a threshold for right-sided histograms.

The unwanted peak in the background entropy $H_{br}$ is a system artefact introduced by the imaging process of particles and can have different reasons. Some were identified as being caused by: (i) higher background flicker as a consequence of high particle densities. A rising number of particles means an increase of speckle noise in the sampling cell due to multiple scattering; (ii) fluctuations in the background that may come from vibrations during the imaging process or instabilities of the light source; or (iii) an inhomogeneous exposure of the camera with a tendency to poorer illumination at the detectors corners—cf. [1].

Figure 6b depicts the same image section of fringe patterns like before but overlaid with the resulting blobs after thresholding. Blue crosses are the centroids of each fringe pattern and correlate to the xy-position of its respective particle unless the shape of the blob deviates too much from an ideal circle of 1. In such cases, the assumption is made that at least two fringe patterns are overlapping and a correction in the count of detected particles is made by +1 for each affected blob. A second cross nearby the actual blob centroid marks the correction made. Of course, the actual particle position does not relate to the determined centroids any more. Orange crosses annotate valid fringe patterns which the algorithm does not recognize. These are missing hits.

### 4.3. DCNN

The detection performance of the DCNN is evaluated in terms of accuracy and precision. A computer with an Intel Xeon W-2145 (Skylake-W) 8-Core CPU and a GPU (NVIDIA GeForce RTX2080 TI) was used. Regarding the training dataset, around 6000 artificially generated holograms have been modelled with the APM and following 64 images were selected as a validation dataset. In these datasets, the number of particles steadily increases from 0 to greater than 200 particles. The number of epochs for the training was 50 and the training time was approximately 4.5 h.

Table 1 shows the accuracy and precision values of the selected samples. In this selection, particles are ranging from 53 to 180. The accuracy is calculated by the number of True Positives + True Negatives divided by the total number of predictions. The formula for precision is the number of True Positives divided by True Positives + False Positives. If no detection of a False Positives takes place, the precision is 1.0.

### 4.4. Comparison of Detection Performance

A quantification of detection performance on real world measurement data is difficult considering that the generation of particles and its supply to the measurement instrument at an unambiguous, steady and reproducable rate is practically impossible. Under these aspects, the actual particle count in the imaged sampling volume of the PIU is in fact unknown and badly verifiable. Therefore, a comparison of particle number concentration $C_{N}$ in numberofparticles/unitvolume = [#/cm${}^{3}$] is most reasonable wherefore the counting results obtained by the three detection methods need to be converted according to [1]:(12)$$C_{N}=\underset{\mathrm{reference}\;\mathrm{CPC}}{\underset{︸}{\frac{N_{\left(\tau \right)}}{Q\cdot \tau_{s}}}}=\underset{\mathrm{PIU}}{\underset{︸}{\frac{N_{\left(V\right)}}{V_{s}}}}$$

The conversion for the PIU is the measured particle count $N_{\left(V\right)}$ over the known sampling volume $V_{s}$ of the PIU. In contrast to that, the reference CPC operates at a known flow rate *Q* and counts the particles $N_{\left(\tau \right)}$ in a certain sampling interval $\tau_{s}$ [33].

In Figure 7, the counting results of the customized HT (top left), the blob detection (top right) and the DCNN (bottom) are compared in terms of the aforementioned particle number concentration. The concentration was ramped from 0 to 2030 #/cm${}^{3}$ which, with respect to image processing, means an average count of roughly 0 to 488 fringe patterns per image. The course of the ideal correlation is drawn as a black reference line. Three frames per measurement point were acquired because the sampling interval of both, the PIU and the reference CPC match best—cf. [1]. The measurement curve is fitted with a polynomial regression function of 3rd order to emphasize the course of detection points.

All three counting methods provide good linearity as long as fringe patterns are spatially well separated ($C_{N}<500$ #/cm${}^{3}$). With increasing particle densities, the likelhood of partially overlapping fringe pattern rises and the detection performance of the DCNN significantly drops. The customized HT and the blob detection can handle particle number concentrations up to approximately C=1250 #/cm${}^{3}$ (or 300 particles per frame) before a regression is noticeable. This implies that partial overlaps are separable very well with these methods (see also in Figure 5 and Figure 6). At even higher counting rates, however, the linear correlation is distorted because of the rising occurence of coinciding particles which is a well known limitation when optically detecting particles [1]. Fringe patterns do not only partially overlap any more but start to superimpose to a mutual pattern of fringes at which neither algorithm is capable of resolving this complex formed patterns (a separation task like this now requires backpropagation algorithms).

Higher error bars in all methods are mostly related to fluctuations of particle rates during the measurement.

#### 4.4.1. Details on Customized HT

From Figure 7, the conclusion can be drawn that the counting performance of the customized HT and the blob detection is very similar in terms of counting rates. The customized HT is more robust against noise and intensity fluctuations in images and therefore less erroneous though. The detectability of fringes at even strong overlaps is very high as was found out in Figure 5. Images without particles are unproblematic and make this method a good candidate for “zero-particle” monitoring.

#### 4.4.2. Details on Blob Detection

Figure 6 illustrates that strong overlaps of fringe patterns lead to merged blobs. This problem is counteracted with the implemented corrective measure where non-circular blobs are treated as multiple occurances. Such blobs are double counted which acts to some extent as a coincidence correction. As a result, the blob detection even gains a slight advantage over the HT at higher particle densities (3) in Figure 7b.

However, annotations (1) and (2) reveal the weaknesses. They indicate sample points where only single measurement frames are distorted by high background fluctuations. As a consequence, the recognition of patterns in these frames fails and yields mainly False Positives, as evident from the concerned measurement frames in Figure 8. Since blob detection is based on histogram thresholding, a misinterpreted threshold leads to incorrect detection hits. The fluctuations add low pixel intensity shares to the histogram which are confused with dark areas of fringe patterns. In the case of zero-particle frames, the impact of a misinterpreted threshold is vast. Because the histogram is divided into foreground and background pixels, zero-particle images are more difficult to classify and prone to misclassifications.

#### 4.4.3. Details on DCNN

At low particle concentrations (<500 #/cm${}^{3}$), the detection performance of the U-Net is comparable with the other methods. The U-Net is capable of classifying zero-particle images as well as images with higher background fluctuations and is as powerful as the HT. The high accuracy at low particle numbers from Table 1 also confirms these good detection rates. At higher concentrations though, Figure 7c with annotation (4) and Table 1 illustrate the strong decrease in detectability. The reason is that, as the number of particle increases, fringe patterns start to overlap for which the network is insufficiently trained. Although the training dataset contained numerous occurrences of fringe pattern overlaps, the network was trained specifically for individual occurrences. Since the accuracy of a DCNN depends on training, an enhanced set of training data would improve the detection performance at least to a limited extent. Due to the lack of real measurement samples and its ground truth data, primarily only modelled holograms could be used. Thus, the training relies on the degree of reality in the Aerosol Particle Model which provided the training data.

Another reason is in the loss of border pixels in every convolution step. Fringe patterns located at the border of images are therefore likely to be missed. Especially at higher particle concentrations, the probability of more particles passing at the edge of the detector increases, degrading the detection performance additionally.

### 4.5. Comparison of Computational Speed

The comparison of computational speed is also based on the dataset of the ramped particle number concentration, also used in Section 4.4. The processing time of all three presented counting methods is examined on every sample point of the measurement curve. These sample points of $C_{N}$ are directly proportional to the counting rate of particles *N* as given in Equation (Equation 12). Figure 9 compares the computational speed of the methods as a function of particle number concentration. It has to be mentioned that both the blob detection and the customized HT are MatLab based algorithms which are executed on CPU without GPU support. The U-Net, on the other hand, is a Python script running on a GPU which is optimized for Artificial Intelligence (AI) applications.

The blob detection and the U-Net are nearly constant over all measurement samples. With an average processing time of roughly 0.45 ms, the blob detection is the fastest method and was selected as the benchmark to which the other algorithms are normalized for comparison. The U-Net is slightly slower, with an average processing time of 0.68 ms or, in terms of computational speed, takes a factor of 1.56 longer in calculation than blob detection.

The customized HT takes at least six times longer and exhibits a strong dependence on the particle rate. There, the number of particles may be interpreted as the number of cycles the algorithm has to iterate to obtain its counting result. It is reasoned in the single transformation of every fringe pattern occurence into Hough space. On closer inspection, the blob detection behaves similarly due to the segmentation and labeling of the growing number of blobs. The impact is at a very small and narrow scale though. Hence, the dependence on the number of particles is negligible.

The U-Net utilizes linear and invariable convolution, pooling and sampling operators and therefore operates at a steady speed.

## 5. Conclusions

The novel application of holography in Optical Particle Counters does not nesseccarily require wavefront reconstruction to reconstruct the particles in the sampled volume. Instead of such typical 3D- backpropagation algorithms, common pattern recognition techniques are sufficient to detect and count interference patterns as valid particles at the Two-Dimensional hologram plane. With a Hough Transform, a variant of blob detection and a Deep Convolutional Neural Network, three different pattern recognition techniques were customized, validated, and compared in terms of detection performance of fringe patterns and computational speed. While model data generated from a holographic Aerosol Particle Model aided the design of the methods, the validation and comparison were based on real measurement samples conducted with the Particle Imaging Unit from [1].

All three methods show basic suitability as counting methods, though with different limitations and drawbacks. At higher particle number concentrations, the rising probability of particle coincidence, as a well known limitation for OPCs, inevitably reduces the detection performance of all methods. Since it takes into account a sort of coincidence correction, blob detection turns out to be the best method with respect to counting rates. The superior computational speed with constant processing times (even without GPU support) enables a Real-Time (RT) application and makes it the best candidate for particle counters. The customized HT is more robust against noise and intensity fluctuations in images and shows slightly better precision at the detection of fringe patterns. However, the longer processing times, which vary as a function of particle rate, disqualify it for practical use. A solution based on a DCNN only works satisfactorily at low particle rates because only a few overlaps of fringe patterns occur. Its accuracy and detection performance may be increased by training with greater datasets and real measurement samples. Although this is a topic for further investigations, similarly high counting rates are difficult to achieve due to the high amount of different overlapping possibilities of fringe patterns. The requirement of a GPU is additionally disadvantageous over CPU-executable RT applications.

The herein provided set of measurement samples spans a range of roughly 0–490 particle counts per measurement frame. Because of coincidence, the used PIU is limited to particle number concentrations of roughly $C_{N}$ = 2000 #/cm${}^{3}$. Further investigations could therefore focus on a redesigned sampling cell to provide a larger detection area for better particle distribution, or smaller fringe patterns at the hologram plane. Both measures would reduce the degree of pattern overlaps and increase the detection performance of all presented methods, or enhance the limit of detectable particles. The latter raises additional research questions with regard to the resolving capabilities of the methods.

## Figures and Tables

**Figure 1 sensors-20-03006-f001:**
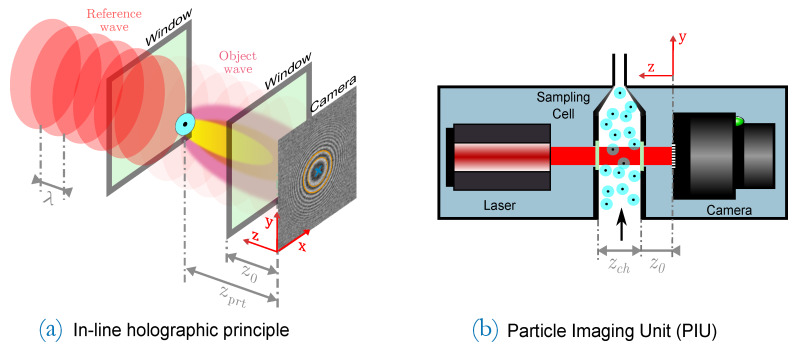
(**a**) in-line holographic principle where a single particle creates a fringe pattern at the camera plane (particles are single neclei in droplets); (**b**) schematic of the in-line holographic counting unit, subsequently called Particle Imaging Unit (PIU) cf. [1].

**Figure 2 sensors-20-03006-f002:**
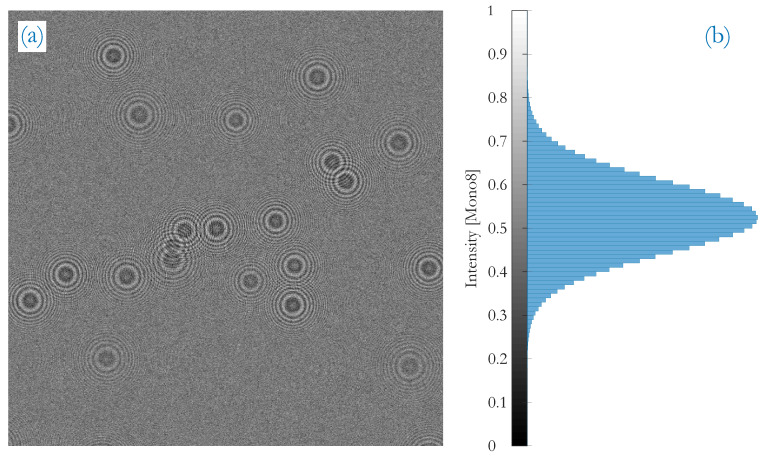
(**a**) example of a typical detection plane at low particle number concentration with overlapping fringe patterns and patterns of different extent as a result of the $z_{prt}$-location in the sampling channel; (**b**) the normalized intensity histogram.

**Figure 3 sensors-20-03006-f003:**
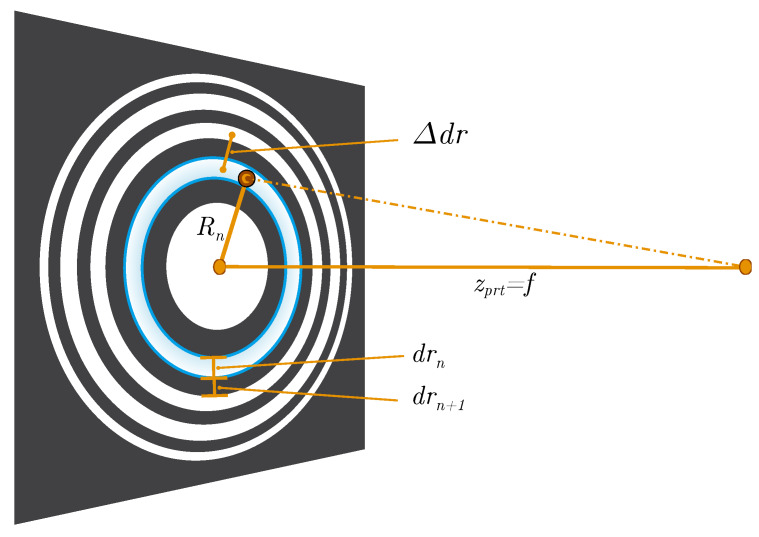
Fresnel Zone Plate (FZP) to estimate the radius $R_{n}$ of fringes and the size of fringe patterns; Δdr is the distance between two successive zone centers of the same parity (even or odd) and may be interpreted as the smallest detail to preserve when lowpass filtering fringe patterns; $dr_{n}$ is the width of *n*th zone.

**Figure 4 sensors-20-03006-f004:**
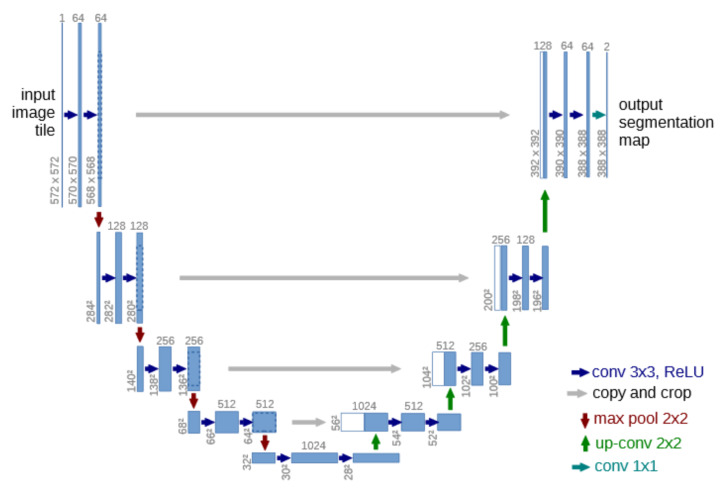
U-Net architecture example for 32 × 32 pixels in the lowest resolution. Each blue box corresponds to a multi-channel feature map. The number of channels is denoted on top of the box. The xy-size is provided at the lower left edge of the box. White boxes represent copied feature maps. The arrows denote the different operations.

**Figure 5 sensors-20-03006-f005:**
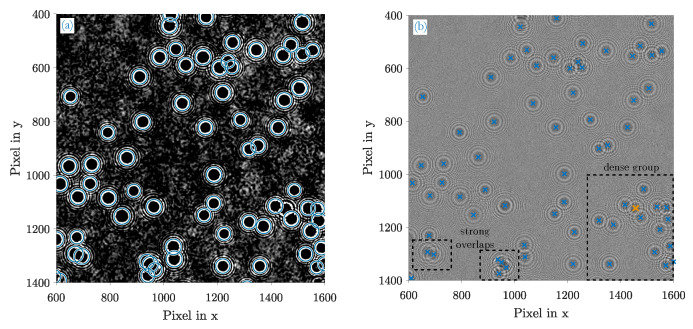
Detection result of the customized HT (selected image section) (**a**) Gaussian filtered fringe patterns ($\sigma_{lp}=2.62$) where all detected fringes are highlighted with circles; (**b**) the original fringe patterns. Its determined centroids equal the actual position of the particles in the xy- plane. There is one missing hit (orange).

**Figure 6 sensors-20-03006-f006:**
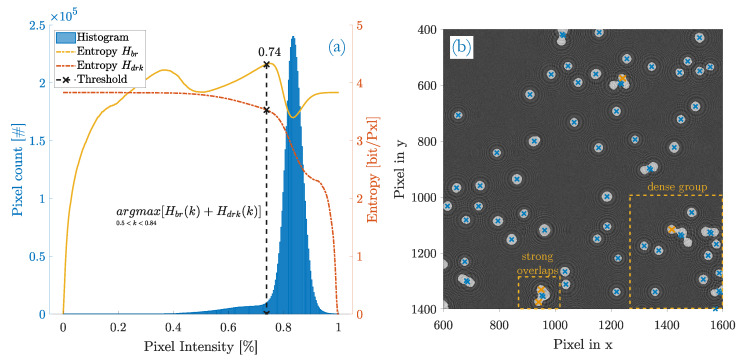
Detection result of blob detection (selected image section). (**a**) histogram and the optimal threshold $k_{opt}$ of the whole image, obtained by maximum entropy thresholding. Equation (Equation 8) needs to be confined to a lower threshold limit set to $L_{1}=0.5$ and an upper limit of $L_{2}=0.84$ which is the peak of the mainlobe; (**b**) fringe patterns overlaid with the corresponding blobs that result from a threshold at $k_{opt}=0.74$. Four hits are missing (orange).

**Figure 7 sensors-20-03006-f007:**
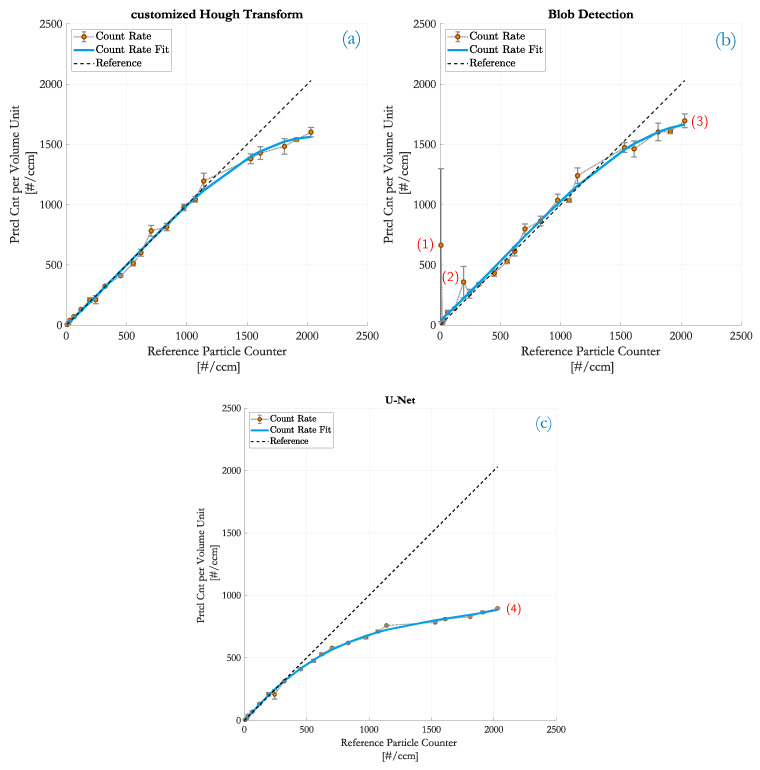
Comparison of the monitored particle number concentration to the counting rates obtained by the PIU. (**a**) customized HT; (**b**) blob detection with maximum entropy thresholding; (**c**) DCNN based on a U-Net.

**Figure 8 sensors-20-03006-f008:**
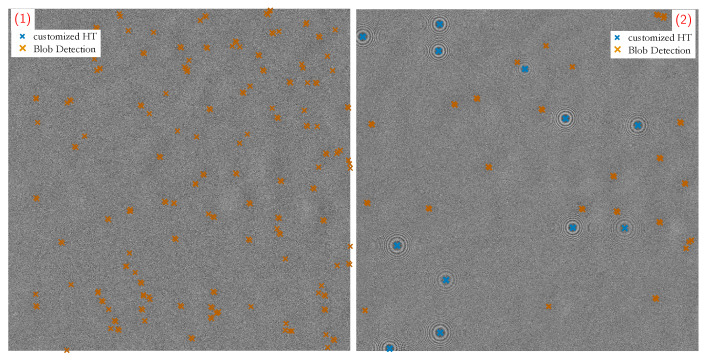
Zoomed segments of measurement samples from Figure 7b that suffer strong background fluctuations: (**1**) zero-particle frame; (**2**) particle number concentration of $C_{N}=194$ #/cm${}^{3}$; while the customized HT outputs correct hits (only True Positives), the blob detection in both scenarios fails (also False Positives) because of a misinterpreted intensity threshold in the histogram.

**Figure 9 sensors-20-03006-f009:**
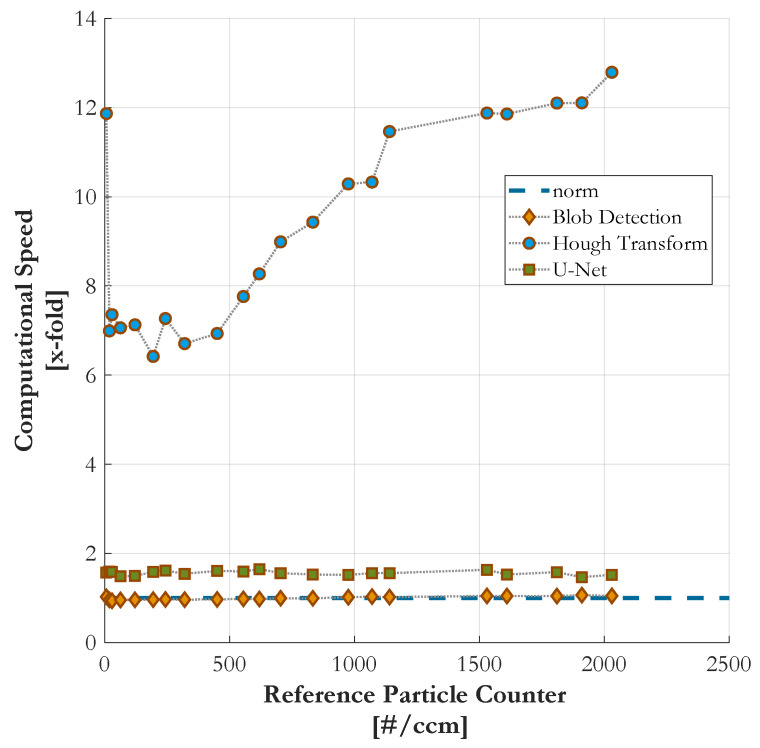
Comparison of computational speed.

**Table 1 sensors-20-03006-t001:** Accuracy and precision of the DCNN.

Number of Particles	Precision	Accuracy
53	0.55	0.98
88	0.45	0.91
103	0.36	0.87
155	0.35	0.74
180	0.25	0.69

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
