# Peer review of "A Comparison of Different Counting Methods for a Holographic Particle Counter: Designs, Validations and Results"

_sensors, 2020, doi:10.3390/s20103006_

Round 1

Reviewer 1 Report

In this paper the authors studied and presented three different techniques applied as Holographic Particle Counters. These techniques are customized for detection and counting of fringe patterns, then the authors compared them in terms of detection performance and computational speed.

The analysis performed would be of interest to widespread applications for which it is important to obtain particle number, thus suitable to be published in Sensors. However, some minor revisions are required. A detailed list with comments and criticisms of the paper is provided below.

  • Figure 1: Since this figure is the same reported in Ref. [1] (open access), it is better to report this reference in the capture.
  • Figure 2: I think in the histogram the intensities are normalized. If yes, please add this clarification in both the text and the capture.
  • Figure 3: What is Δrres?
  • 5 line 132: In figure 3 there isn't Δdr, please clarify.
  • 6 lines 141-142: Why these values are chosen? Can the authors add some comments on this choice?
  • 11 line 298: there is a typo (missing).
  • Figure 7 top right: Why at concentration (2) the procedure fails while at lower concentrations it works well? Can the author clarify this?
  • In the conclusion a critical point of view of the authors, such as future perspectives or suggestions for further investigations are missing.
  • For better readability, I recommend to insert letters in figures and do not refer to them as right, left, top and bottom. For example, Figure 6 left and right could become Figure 6 (a) and (b).

Reviewer 2 Report

The authors report, in this work, a novel method used for particle identification using digital inline holography. This work is well written and presented appropriately. It presents novel results with a concise and comprehensive manner and I suggest that it is accepted almost as is.

It is unlikely that the system will replace CPC or OPC soon. However, both systems suffer from several limitations that the HIC may be superior. This is related to a topic of great interest nowadays in the field of aerosol science, that of giant particles. Under that perspective, I would ask to authors to write a small paragraph related to the limitations of their system.

It is clear that the method the authors describe can be applied only to spherical particles. What are the limitations though? Dry sea salt particles are cubes with spherical edges. Because they come with other inorganic and organic particles they end up closer to the spherical shape and in the sizes the authors investigate in this work. Would that work? What are the sphericity limitations?

What is the lower and upper (if any?) size limit of detection. An example being nucleation events. In this case, the size of freshly  nucleated particles  after grown in the CPC’s saturator will be a function of the original size. What would be the expected discrepancy in that case?

Taking into account that this work is to a very large extent computational, I encourage the authors to submit a persistent link of their detection algorithms for transparency. This is by no means mandatory.

Minor comments

Please mention the working fluid when you refer to particle growth by condensation in Lines 35-38. Butanol, water, DEG, isopropanol?

Line 398: bob should be blob
